⊙ | **Open Peer Review** | Host-Microbial Interactions | Research Article

# Increased fungal burden in the gastrointestinal tract of brain-dead organ donors

Erika J. Nachman,[1] Colleen K. Ardis,[1] A. Kyle B. Ardis,[1] Javier Nieto,[2] Madeline M. Bresson,[1] Clare M. Robertson,[1] Maggie N. Seale,[1] Nora M. Villafuerte,[1] Zhe Lyu,[1] Eva C. Preisner,[1] Heather A. Danhof,[1] Sara C. Di Rienzi,[1] Yolanda T. Becker,[2] Robert A. Britton[1]

**ABSTRACT**    Approximately 3% of organ transplant recipients will be diagnosed with an invasive fungal infection (IFI), with 10%–47% being fatal. The source of IFI is typically not known, but *Candida* spp. cause the majority of IFI cases. We obtained intestines from two types of organ donors, brain-dead (BD) ($n = 7$) and donation after circulatory death (DCD) ($n = 5$), to sample microbes from the mucosal surface and the lumen. BD donors were parenterally fed and treated with corticosteroids in the days before procurement, whereas DCD donors were enterally fed and not given corticosteroids. We found fungi present in 81% of BD donor samples compared to 31% of DCD donor samples. The fungal load was significantly increased in most of the intestinal sites in BD donor samples over DCD donor samples. We recovered 12 isolates of *Candida albicans*, 10 isolates of *Nakaseomyces glabratus,* and one isolate of *Candida tropicalis* from the human intestine of nine organ donors. Sequence analysis revealed a high prevalence of gene families associated with virulence and drug resistance, the latter of which was corroborated by antifungal resistance testing. Overall, this study provides evidence of increased fungal presence in BD donors compared to DCD donors, which suggests that corticosteroid administration and parenteral feeding strategies of BD donors may impact the burden of IFI in transplant recipients.

**IMPORTANCE**  Invasive fungal infections pose a significant risk to organ donor recipients, and the sources of these fungal infections are mostly unknown. Our study investigated fungal differences in the gastrointestinal tract between two types of organ donors: brain-dead and donation by circulatory death, for the first time. We found that brain-dead donors had a significant increase in the frequency and abundance of fungi throughout the gastrointestinal tract. We speculate that differences in care, such as corticosteroid administration and feeding methods before organ procurement, may begin to explain the difference in fungal presence between the donors. We propose further studies into the administration of steroids and delivery of nutrients to organ donors before procurement to impact fungal load and reduce life-threatening invasive fungal infections in the recipients.

**KEYWORDS**    fungi, transplant infectious diseases, *Candida*, intestinal colonization

Solid organ transplants (SOTs) of the kidney, lungs, heart, liver, and small intestine (SI) are life-saving procedures. In 2023, 44,815 organ transplants were performed in the US (1). Invasive fungal infections (IFIs) are a serious complication for SOT recipients, resulting in an invasion of fungi in deep tissue and organs (2). The occurrence of IFI in SOT is 1.3%–11.6% (2, 3), with mortality rates ranging from 20% to 40% (3). IFIs can spread systemically, including the blood, heart, and bones, causing further complications to the SOT recipients (2, 4).

**Peer Reviewer** Cornelius J. Clancy, University of Pittsburgh School of Medicine, Pittsburgh, Pennsylvania, USA

Address correspondence to Robert A. Britton, Robert.britton@bcm.edu, or Yolanda T. Becker, ybecker@lifegift.org.

The authors declare no conflict of interest.

*[This article was published on 18 June 2025 with errors in the text. The errors were corrected in the current version, posted on 9 July 2025.]*

The source of IFI in SOT recipients is usually unknown, but it is suspected to come from the environment or contamination from the donor organ (4). In the recipients, *Candida* spp. are the most common culprit of IFI in all organs except the lungs (5), and infections occur within the first few weeks to months after transplantation (4). While direct sources of IFI are not typically identified, in one case, *Candida* spp. were found as a contaminant in the organ fluid preservation solution (6). *Candida* spp. inhabit human mucosal sites as opportunistic pathogens, and the World Health Organization prioritized *Candida albicans* and *Candida auris* as critical pathogens, contributing to increased hospitalizations, drug resistance, and difficulty in treating IFI (7).

Fungi comprise 0.01%–0.1% of the overall microbial abundance of a healthy gastrointestinal tract (GIT) consisting primarily of *Candida*, *Saccharomyces*, and *Malassezia* species (8). In a prospective human cohort study, an expansion of the gastrointestinal tract (GIT) *Candida* spp. preceded systemic fungal infections (9). Furthermore, SI transplant recipients are diagnosed with IFI after transplant at a much higher rate (11.6%) than recipients of other organs (4). This suggests that either the procedure for recovery of the SI is prone to fungal contamination or that IFI-causing fungi may originate from the SI. One possibility for why the SI may have more fungi in organ donors can be gleaned from the total parenteral nutrition (TPN) literature. In contrast to enteral feeding, TPN patients receive nutrients intravenously (IV), bypassing the GIT. TPN patients have an increased risk of candidemia compared to patients who are enterally fed (OR = 3.92) (10). Brain-dead (BD) donors receive TPN, often for multiple days before organ procurement, suggesting a potential link between feeding strategies in organ donors and fungal overgrowth.

The rate of IFI in donation after circulatory death (DCD) donors has not been widely studied, with one study reporting a slight decrease in IFI rate from DCD kidney donors (1.2%) compared to the reported 1.3% rate in BD donors (4, 11). While differences in organ health and transplantation success rates between DCD and BD have been assessed, the role of enteral feeding, donor type, and the donor's intestinal microbiome has not yet been explored (12). We hypothesized that donor care before procurement could impact the GIT fungal burden. To investigate this question, we compared a small cohort of two types of organ donors: brain-dead (*n* = 7) and donation after circulatory death (*n* = 5). We investigated the mycobiome throughout the GIT by sequencing-based and culture-dependent approaches. We found major differences in care before procurement, where DCD donors were enterally fed significantly closer to procurement and were not given corticosteroids. BD donors, however, were fed parenterally in the days prior to procurement and received corticosteroids. We observed an overgrowth of fungi in the GIT of BD organ donors compared to DCD donors. Our observations suggest that corticosteroid usage and feeding practices prior to organ procurement may contribute to alterations in fungal load in the gut. Further investigation will be needed to determine if exposure to enteral feeding may reduce the risk of IFI in SOT recipients.

## MATERIALS AND METHODS

### Donor inclusion and exclusion criteria

GITs were collected between 6 February 2021 and 19 June 2024. To identify organ donors that would best represent an adult GIT microbiome, our inclusion criteria were donor age ≥18 years old, located in the Texas Medical Center, no known bowel disease, surgery, or trauma (inflammatory bowel disease, *Clostridioides difficile* infection), hospitalized for less than 28 days, and BD for less than 7 days. We accepted both BD and DCD donors. DCD donors had a cutoff time of 90 minutes after the removal of life-support machinery for the inclusion of the organs in this study.

The exclusion criteria included patients with positive serological tests for hepatitis B, hepatitis C, or HIV, significant GIT bleeding during hospitalization, prior intestinal resection, a medical examiner case, transferred to the hospital from a long-term care

facility, cases with GIT tissue recovery following organ recovery, or our lab being unable to process the specimen.

## Organ dissection

The entire length of the GIT from the stomach to the sigmoid colon (when available) was surgically removed from eligible donors and placed onto ice for transportation to the laboratory within 45 minutes of removal. The intestines were anatomically sectioned into 13 parts consisting of the stomach (fundus, body, and pylorus), duodenum, jejunum, (upper, middle, and lower), ileum (upper, middle, and lower), ascending colon, transverse colon, descending colon, and sigmoidal colon. (Fig. 1). Sections were bound tightly with cotton string to prevent leaking or mixing of contents between segments during dissection. Unless otherwise noted, all procedures were performed within the standard laboratory environment (atmospheric oxygen conditions and room temperature).

## Luminal sample collection

A volume of 30 mL of sterile, room temperature, anoxic phosphate-buffered saline (PBS) was injected into closed-off tissue sections with a sterile 16-gauge needle, mixing with luminal contents. The PBS was then removed from the tissue by syringe, transferred to conical tubes, and delivered to an anoxic chamber (5% $H_2$, 5% $CO_2$, 90% $N_2$, and 0–50 ppm of $O_2$). One milliliter aliquots in triplicate were flash frozen in liquid nitrogen for future quantitative real-time polymerase chain reaction (qRT-PCR) analysis. An equal volume of anoxic 30% glycerol-PBS was added to 20–50 mL of luminal contents for cultivation analysis in anoxic conditions. The samples were then homogenized by agitation on a vortex mixer for 30–90 seconds, subsequently removed from the anoxic chamber, flash-frozen in liquid nitrogen, and stored at −80°C. Large intestine luminal samples were collected by the same process, except when contents were solid, the fecal matter was transferred into a 50 mL conical tube with a sterile scoop and processed as described above.

## Luminal sample processing

Luminal samples were thawed at room temperature under anoxic conditions and resuspended using a vortex mixer. Samples were centrifuged at $200 \times g$ for 2 minutes at 4°C to pellet large debris. All 4°C centrifugation was completed with the Thermo Fisher Scientific Heraeus Multifuge X3R Benchtop Centrifuge with a 75003180 rotor (Waltham, MA, USA). The supernatant was transferred to a new, pre-weighed conical tube in the anoxic chamber with a sterile serological pipette. Then, the luminal contents were centrifuged at $2,800 \times g$ for 10 minutes at 4°C. After the removal of the supernatant, the cell pellets were weighed. A volume of 2.5 mL of anoxic PBS per 1 gram of cell pellet was added to each sample and homogenized for 10 seconds in anoxic conditions. An equal volume of 30% glycerol-PBS was added to cryovials for stocks of microbial communities. The samples were then homogenized to combine the glycerol-PBS, and 1.8 mL was aliquoted into cryovials for storage at −80°C.

Solid large intestine samples were thawed in anoxic conditions, and any insoluble food residues were removed using a sterile wooden stick. A maximum of 6 g of sample was transferred to a pre-weighed 50 mL conical tube. Twenty-four milliliters of anoxic PBS was added to the sample. Outside the anoxic chamber, the sample was shaken at 1,500 rpm for 10 minutes at 4°C and then separated by centrifugation at $200 \times g$ for 4 minutes at 4°C. The sample was immediately returned to the anoxic chamber. Twenty milliliters of the supernatant was transferred to a 50 mL Falcon tube, and an equal volume of sterile 30% glycerol-PBS was added for a final sample concentration of 15% (vol/vol) glycerol-PBS. The sample was briefly vortexed before aliquoting into cryovials for storage at −80°C or plated for culturing that day (see below).

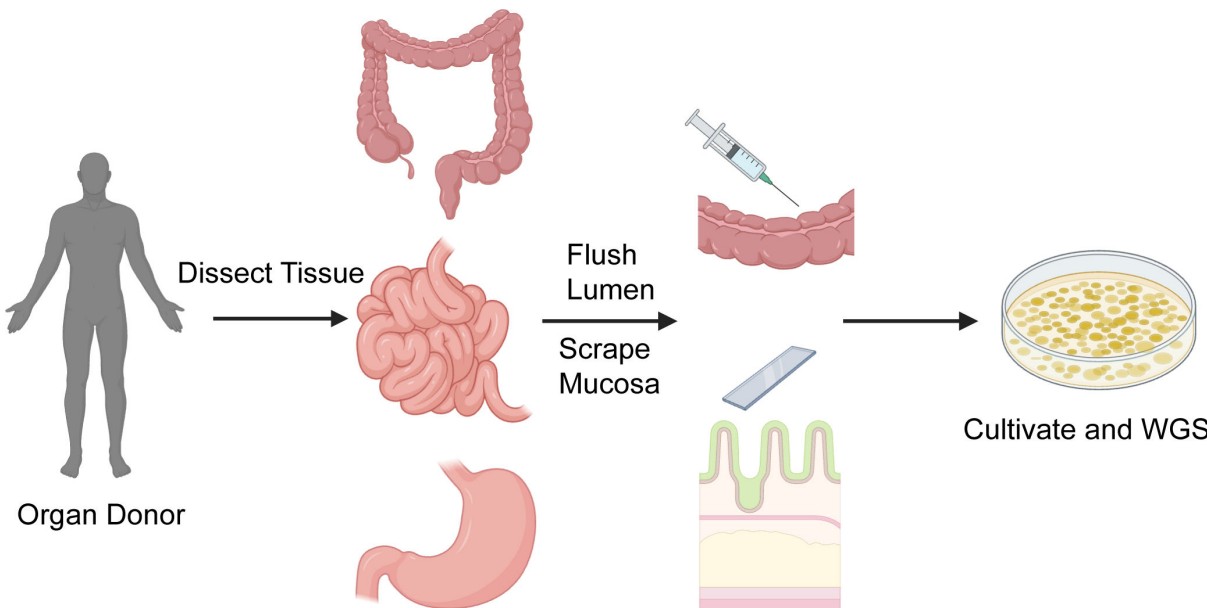

**FIG 1** Workflow for the collection of gastrointestinal fungal samples from organ donors. GIT arrived at the lab within 45 minutes of removal from the donor. The intestines were dissected into anatomical parts (e.g., stomach, duodenum, middle jejunum, middle ileum, transverse colon). Samples for cultivation were obtained from luminal flushes and mucosal scrapings. The resulting fungal growth was purified and sent for whole-genome sequencing for identification. Created in BioRender. https://BioRender.com/r58r358.

## Mucosal collection and processing

Immediately following the collection of luminal contents, each tied-off intestinal region was cut open, and the tissue was placed in a PBS bath in a shallow metal container to remove residual luminal contents. The tissue was then scraped using a sterile, glass microscope slide by holding one end of the tissue and sliding the glass slide across the tissue to detach the mucus. Mucosal scrapings were transferred into cryovials and a pre-weighed 15 or 50 mL conical tube containing 4.5 mm sterile glass beads filled to the 3 mL mark. The cryovials were immediately flash-frozen in dry ice for future sequencing. Cultivation-dependent samples were derived from the mucus placed into the conical tubes. For these samples, the net weight of the conical tube with mucus was calculated, and 2 mL of PBS per 1 g of mucus with 5–25 sterile beads, depending on the volume of mucus, were added to the sample to disassociate the microbes. The samples were then homogenized using a vortex at 4°C for 1 minute to break up the mucus. The mucosal samples were transferred into an anoxic chamber, where the mucus was transferred into a new conical tube. Sterile 30% glycerol-anoxic-PBS was mixed using a vortex with the sample and stored at −80°C.

## Plating and isolation of fungi

Yeast-peptone-dextrose (YPD) plates were made with 10 g/L of yeast extract (Fisher, Pittsburgh, PA, USA, BP1422), 20 g/L each of peptone (Oxoid, Thermo Fisher Scientific, Waltham MA, USA, LP0072B), 20 g/L of dextrose anhydrous (Avantor, Radnor, PA, USA, BDH9230), and 15 g/L of agar (BD Difco, Franklin Lakes, NJ, USA, 214010). After autoclaving for 30 minutes at 121°C under 15 psi and cooling to 55°C, 0.1 and 0.2 g/L of vancomycin and gentamicin (Avantor, Radnor PA, USA, 1405-41-0 and 97062-554), respectively, were added to reduce the bacterial load. Luminal and mucosal samples from the stomach, duodenum, middle jejunum, middle ileum, and either ascending or transverse colon were plated undiluted, $10^{-2}$, and $10^{-3}$ dilutions in anoxic PBS in duplicate. The plates incubated at 37°C in atmospheric $O_2$ for 24 hours. This resulted in 104 samples from 6 BD donors across 5 intestinal sites in duplicates (52 luminal and 52

mucosal) and 90 samples from 5 DCD donors across 5 intestinal sites in duplicates (48 luminal and 42 mucosal). However, we did not have colon luminal samples from donors 8 (BD), 10 (DCD), and 13 (BD). From the mucosal samples, we did not have the stomach, duodenum, jejunum, and colon from donors 9 (BD), the fundus from 6 (DCD), and the stomach, jejunum, and colon from 10 (DCD). This resulted in the final 104 samples from BD donors and 90 samples from DCD donors. Colonies were counted, and single colonies were purified by streaking and underwent whole-genome sequencing. Frozen stocks were made by growing the isolates in YPD culture overnight at 37°C with shaking at 200 rpm. Glycerol-PBS (30%) was added in equal volume to the culture and stored at −80°C. Donor 1 was excluded from plating due to probable bacterial contamination of processed samples, but one fungal isolate was included in whole-genome sequencing.

## Fungal and bacterial DNA isolation

Fungal isolates were acquired from seven BD donors and two DCD donors. Fungal DNA was isolated using the lithium acetate method and underwent whole-genome sequencing (13). Bacterial and fungal DNA from luminal and mucosal scrapings wasextracted using the DNeasy PowerSoil Kit Pro for qRT-PCR (Qiagen, Aarhus, Denmark, ID 47014). DNA was extracted according to the manufacturer's protocol with two 30-second bead-beating steps. The extracted DNA was eluted in 30–50 µL of the provided elution buffer.

## ITS2 and quantitative real-time polymerase chain reaction sequencing of communities

ITS2 sequencing and qRT-PCR quantification were performed by the Alkek Center for Metagenomics and Microbiome Research at Baylor College of Medicine, Houston, TX, USA. The fungal load was quantified using FungiQuant (14), a qRT-PCR assay that measures ITS2 copies using the following primers: forward 5′-GGRAAACTCACCAGGTC-CAG-3′ and reverse 5′-GSWCTATCCCCAKCACGA-3′ (see supplemental material). Bacterial load was quantified using 16S rDNA qRT-PCR (forward 5′ CGGTGAATACGTTCYCGG3′ and reverse 5′ GGWTACCTTGTTACGACTT3′), and the copies/ng of DNA were calculated using a standard curve for each of the primer sets, ranging from $10^1$ to $10^7$ copies. The QuantStudio DX machine was used for all qPCR quantifications.

## Assembly of whole-genome sequencing and annotation of fungal isolates

Whole-genome sequencing of fungal isolates was completed by SeqCenter (Pittsburgh, PA, USA). Genomes were sequenced using an Illumina NovaSeq X Plus sequencer to produce 2 × 151 bp paired-end reads. The reads were demultiplexed, quality checked, and adapters were trimmed using bcl-convert (version 4.2.4) (15). Genomes were assembled using SPAdes (16). Bin quality metrics such as completeness, contamination, and $N_{50}$ were evaluated using CheckM and QUAST (17, 18) (version 5.0.2). Taxonomy was assessed with Kaiju's nr_euk database (19) (1.10.1). Gene prediction was completed for the subset of genes analyzed in this study by creating a local blast database for each species (20). The tblastn command was used to align proteins to the translated fungal genome. Positive hits were called based on the length of amino acid coverage (60% or more of the entire protein) and amino acid positivity similarity (50% or more). All bioinformatics work was conducted on a SLURM-based cluster managed by the Biostatistics and Informatics Shared Resource, supported by NCI P30-CA125123 and institutional funds from the Dan L. Duncan Comprehensive Cancer Center and Baylor College of Medicine. The presence of virulence factors and antibiotic resistance genes was visualized using the ComplexHeatmap (version 2.20.0) R package (version 4.1.1). The clustering was calculated using the Jaccard method for the isolates (columns) and the Euclidean method for the genes (rows).

## Resistance to antifungal drugs

Fungal isolates were streaked onto YPD agar and incubated overnight at 37°C. The resulting cultures were then diluted to $OD_{600}$ of 0.09, and 100 µL of culture was spread-plated on RPMI agar plates (Gibco, Thermo Scientific, #31800022, Waltham, MA, USA) supplemented with 2% dextrose anhydrous and 3.5% MOPS (Millipore, M1254, Burlington, MA, USA), using sterile beads. After the culture dried for 30 minutes, Liofilchem minimum inhibitory concentration (MIC) test strips for micafungin, amphotericin B, caspofungin, and fluconazole (Fisher Scientific, 22-778-050, 22-777-981, 22-777-984, and 22-777-963, Waltham, MA, USA) were sterilely placed in the center of the plate and incubated at 37°C for 24 hours. This was repeated for three individual overnight cultures per isolate, and MICs were recorded.

## Statistical analysis

Statistical analysis was evaluated using R (version 4.1.1). The Mann-Whitney $U$ test was used to determine the statistical difference in age and last known oral feeding between the donors (Table 1). Fisher's exact test was used for statistical differences between sex and ethnicity/race between the donor types (Table 1). Similarly, Fisher's exact test with Benjamini-Hochberg correction was used to determine the significance of the presence of fungi (Tables 3 and 4). Mann-Whitney $U$ tests with Holm's correction were used to determine the statistical significance of fungal burden across the GIT (Fig. 2). For qRT-qPCR data, statistical significance was determined by a mixed linear model with donor type (BD or DCD) as a fixed effect and donor ID as a grouping variable to control for repeated measures (Fig. S1). The data were log10-transformed for 16S rRNA qRT-qPCR data for statistical analysis.

## RESULTS

### Timing of enteral feeding differs between BD donors and DCD donors

We received GIT organ donations from 11 donors (5 males and 6 females) between 23 and 54 years old and of multiple races and ethnicities (Table 1). Extended demographics on the donors can be found in Table S1. There was no significant difference between donor type and age (Mann-Whitney $U$, $P = 0.23$), sex, and race and/or ethnicity (Fisher's exact test $P = 0.57$ and $P = 0.06$, respectively) (Table 1). However, DCD donors were enterally fed significantly closer to procurement than BD donors (14.4 vs 92 hours, Mann-Whitney $U$, $P = 0.007$) (Table 1).

**TABLE 1** Demographics of organ donors

| Donor | Sex | Age (years) | Race or ethnicity | Type of donation | Last known enteral feeding prior to procurement (hours)[a] |
|---|---|---|---|---|---|
| 2 | Female | 34 | White | BD | 72 |
| 3 | Male | 23 | Hispanic | BD | 72 |
| 4 | Female | 25 | White | BD | 72 |
| 8 | Female | 30 | Hispanic | BD | 96 |
| 9 | Male | 45 | Hispanic | BD | 96 |
| 13 | Female | 41 | Hispanic | BD | 144 |
| 5 | Male | 23 | African American | DCD | 6 |
| 6 | Male | 54 | White | DCD | 32 |
| 7 | Female | 45 | African American | DCD | 14 |
| 10 | Female | 48 | White | DCD | 12 |
| 12 | Male | 40 | White | DCD | 8 |

[a]Significance between BD and DCD samples for last known oral feeding before procurement ($P$-value = 0.007, Mann-Whitney $U$).

## Only brain-dead organ donors were administered steroids prior to procurement

In addition to differences in nutritional delivery, all BD donors received at least one dose of hydrocortisone before procurement, while DCD donors did not. This is the standard of care for BD donors to prevent injury to the organs due to inflammation following brain death (Table 2). All donors received at least one intravenous antibiotic dose prior to organ recovery (Table 2) with piperacillin/tazobactam (73%) and vancomycin (64%) administered most frequently (Table 2). While BD donors were on antibiotics (3.3 days) longer on average than DCD donors (1.6 days) (Table 2), we did not see any difference in bacterial load by qPCR (Fig. S1). Only two donors (donors 3 and 13) received one dose of an antifungal, micafungin.

## Brain-dead donor samples have increased fungal presence

To collect microbes along the length of the GIT, luminal and mucosal samples were collected and cultivated on agar plates (Fig. 1). The resulting isolates were purified and underwent whole-genome sequencing (Fig. 1).

To quantify the number of fungi in each section of the intestine from 11 total donors, we plated intestinal samples on YPD plates containing antibiotics to suppress bacterial growth. Fungal frequency was reported by the presence or absence of fungal colony-forming units (CFUs). We observed significantly higher viable fungal presence across BD GIT samples (81%) from all six donors tested, while only 31% in DCD samples, driven mainly by two DCD donors 6 and 10 (Table 3). Similarly, analysis by intestinal region revealed that BD donor samples had a higher frequency of viable fungi (82%, 89%, 82%, and 100%) in the stomach, duodenum, jejunum, and colon compared to DCD donors (31%, 30%, 22%, and 29%) except for the ileum (67% in BD and 40% DCD) (Table 4).

We next quantified the fungal load of the cultivated samples. We observed that CFUs were significantly increased in both luminal and mucosal samples from all intestinal segments of BD donors compared to DCD donors, except in the ileum. Mean CFU differences ranged between 100 and 10,000 times higher in luminal samples (Fig. 2A) and 10 and 100 times higher in mucosal samples (Fig. 2B). Significant differences between donor groups were not observed in the ileum, possibly due to the lower number of positive BD fungal samples (Table 4; Fig. 2A and B).

To corroborate this finding, the fungal copy number was measured by qRT-PCR of the ribosomal DNA ITS2 region to measure fungal burden. We found that BD donors had a significant increase in the average fungal copy number (494 copies/ng) compared to DCD donors (12 copies/ng), particularly in the mucosal samples (Fig. S1A). However, there was no difference in the ITS2 signal in the luminal samples (Fig. S1B). Furthermore, we noted that the DCD samples without cultivable fungi also had lower ITS2 fungal signal, consistent with decreased fungal presence (Fig. S1A and B; Table 4). There was no significant difference in bacterial burden across donor types (Fig. S1C and D).

## Selected fungal isolates are resistant to antifungals

Drug resistance is a growing issue for the treatment of fungal infections. We performed whole-genome sequencing analysis of a select group of 23 fungal isolates to gain further insights into the drug-resistant potential of a subset of fungi present in the organ donors. There were 18 isolates representing BD donors and 5 isolates isolated from the two DCD donors with detectable fungi. These isolates were chosen to prioritize at least one strain from any donor with a positive fungal signal. Therefore, multiple isolates of BD donors 1, 3, 8, and 9 were included. As donors 4 and 13 had less consistent recovery from all sites of the GI, only one isolate from each donor was included. Finally, two isolates from each DCD donor with a positive fungal signal were prioritized. Out of the 23 isolates analyzed, 12 were *Candida albicans*, while 10 were *Nakaseomyces glabratus*, and 1 was *Candida tropicalis*. Of the DCD donors, donor 6 had two *N. glabratus* isolates and one *C. albicans* isolate, while donor 10 had two *C. albicans* isolates.

**TABLE 2** Antibiotic, antifungals, and steroid treatment for organ donors[a]

| Donor ID | Antibiotics/antifungals | | | Steroids | | |
|---|---|---|---|---|---|---|
| | Name | Dose (g, hours) | Time (days) | Name | Dose (g, interval) | Time (days) |
| 2 | Piperacillin/tazobactam | 3.4, 8 | 2.6 | Hydrocortisone | 0.13, single dose | 0 |
| | Vancomycin | 1, 12 | 1.6 | Hydrocortisone | 0.05, 6 hours | 1.7 |
| 3 | Piperacillin/tazobactam | 3.4, 8 | 1.9 | Hydrocortisone | 0.05, 6 hours | 2.0 |
| | Vancomycin | 1, 12 | 4.8 | | | |
| | Metronidazole | 0.5, 12 | 6.7 | | | |
| | Micafungin | 0.1, 24 | 3 | | | |
| 4 | Vancomycin | 1.5, 12 | 3.2 | Hydrocortisone | 0.05, 6 hours | 3.0 |
| | Piperacillin/tazobactam | 3.4, 6 | 3.6 | | | |
| 8 | Vancomycin | 1, 12 | 1.2 | Hydrocortisone | 0.3, single dose | 0 |
| | Piperacillin/tazobactam | 3.4, 8 | 2.8 | Hydrocortisone | 0.1, 8 hours | 2.8 |
| 9 | Piperacillin/tazobactam | 3.4, 8 | 2.3 | Hydrocortisone | 0.1, 8 hours | 2.4 |
| | Cefepime | 1, 12 | 6 | | | |
| 13 | Vancomycin | 1, single dose | 0 | Hydrocortisone | 0.3, single dose | 0 |
| | Vancomycin | 1.3, 12 | 1.5 | | | |
| | Levaquin | 0.75, 24 | 2.4 | | | |
| | Piperacillin/tazobactam | 3.4, 8 | 6 | Hydrocortisone | 0.1, 8 hours | 3.1 |
| | Ceftriaxone | 1, single dose | 0 | | | |
| | Micafungin | 0.1, single dose | 0 | | | |
| 5 | Piperacillin/tazobactam | 3.4, 8 | 1.8 | None | | |
| 6 | Piperacillin/tazobactam | 3.4, 8 | 1.3 | None | | |
| 7 | Piperacillin/tazobactam | 3.4, 8 | 2.2 | None | | |
| | Ceftriaxone | 1, 24 | 1.8 | | | |
| 10 | Cefepime | 1, 8 | 2.7 | None | | |
| | Vancomycin | 1, 12 | 1.7 | | | |
| 12 | Cefepime | 2, 8 | 0.5 | None | | |
| | Vancomycin | 1, 24 | 0.6 | | | |

[a]Time indicates the total treatment interval.

We tested the susceptibility of our fungal isolates to clinical antifungal drugs *in vitro* using MIC test strips. A representative group of fungal isolates was tested for resistance against common antifungals: fluconazole, caspofungin, micafungin, and amphotericin B. Using CLSI breakpoint guidelines for *C. albicans* and *N. glabratus*, we found that 100% of isolates tested were resistant to fluconazole, including the control strain *C. albicans* SC5314 (Table 5). The majority of isolates were susceptible to antifungals except for *N. glabratus* from donor 6, which was resistant to caspofungin (Table 5). While there is no standard breakpoint for amphotericin B for these fungal species, isolates had varying susceptibility to amphotericin B, ranging from 0.06 to 0.42 µg/mL. Five isolates had higher average susceptibility to amphotericin B than to control strain *C. albicans* SC5314 (0.13 µg/mL) (Table 5). Overall, while all the isolates were resistant to fluconazole, only one isolate was resistant to the primary and secondary lines of IFI treatment.

To corroborate our phenotypic findings, we mined the isolate genomes for drug resistance-related genes (Table S2). *FLU1*, *CDR1*, *ADA2*, *CUP9*, *SNQ2*, *FCR1*, and have

**TABLE 3** Higher overall fungal frequency in BD donor samples by cultivation[a]

| Donor | No. of samples | % positive | *P* value[b] |
|---|---|---|---|
| BD | 104 | 81 | *P* < 0.0001 |
| DCD | 90 | 31 | |

[a]Fundus, duodenum, jejunum, ileum, and ascending colon luminal and mucosal samples from BD and DCD donors were plated, when available, for viable fungal growth in duplicate. The presence or absence of fungal growth was recorded by percent positive, summarized as BD or DCD positive/total. Those ratios are displayed as a percent of BD or DCD positive.
[b]Statistical significance was measured by Fisher's exact test.

**TABLE 4** Higher fungal frequency across the GIT sites of BD donors by cultivation[a]

| Site | BD positive/ total (n = 6) | BD % positive | DCD positive/ total (n = 5) | DCD % positive | P value[b] |
|---|---|---|---|---|---|
| Stomach | 18/22 | 82 | 5/16 | 31 | P = 0.004 |
| Duodenum | 16/18 | 89 | 6/20 | 30 | P = 0.005 |
| Jejunum | 18/22 | 82 | 4/18 | 22 | P = 0.001 |
| Ileum | 16/24 | 67 | 8/20 | 40 | P = 0.222 |
| Colon | 16/16 | 100 | 4/15 | 29 | P < 0.001 |

[a]BD and DCD donor samples were compared for each intestinal site for fungal frequency.
[b]Statistical significance values were derived from Fisher's exact test with a Benjamini-Hochberg multiple comparison correction.

been reported to impact antifungal resistance (Table S2) (21–26). We observed that all isolates had three or more drug-resistance-related genes (Fig. S2) and suggested a strong correlation between *FLU1, ADA2, SNQ2,* and *CDR1* presence and fluconazole resistance.

## Fungal isolates harbor expected virulence genes

As opportunistic pathogens, *N. glabratus* and *Candida* spp. have multiple well-characterized virulence genes. To assess the pathogenic potential, we interrogated the presence of virulence genes of *Candida* spp. (Fig. S3A) and *N. glabratus* (Fig. S3B) in the genomes of each isolate. Among the key characteristics of virulent fungi are survival within immune cells, adherence to epithelial cells, dimorphic switching from yeast to hyphal form, and secretion of enzymes and a toxin to facilitate tissue invasion (Table S3). We focused on invasion (*SSA1*) (27), adhesions (*AWP1*) (28), and survival in macrophages (*SKN7*, *YAP1*, *MSN4,VPS34, HOG1*) (29–31) for *N. glabratus* isolates (Table S3). For *Candida* spp., important virulence factors include adhesion factors to bind to host cells (*HWP1, ALS1*, and *ALS3*) (32–34), a lytic peptide toxin candidalysin (*ECE1*) (35), secreted aspartyl proteases (*SAP4* and *SAP5*) (36), a lipase (*LIP8*) (37), and a morphogenesis regulator (*EFG1*) (38) (Table S3).

We detected most of the expected genes in *Candida* spp. and *N. glabratus,* maintaining their status as potential threats if they escape the intestinal tract (Fig. S3). However, we detected variability in adhesion genes (*AWP1*, *HWP1*, and *ALS1*) across the three species (Fig. S3). While there have been reported variations in *ALS1* presence in

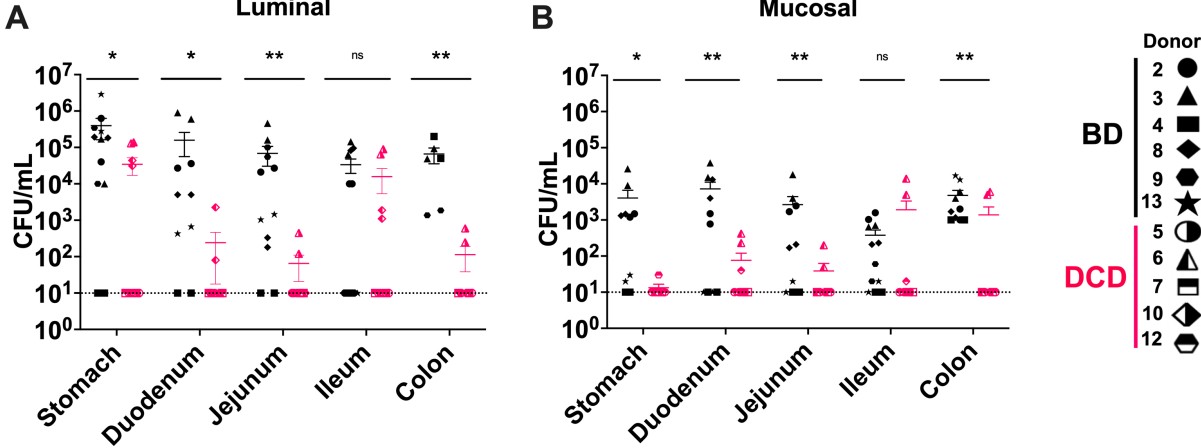

**FIG 2** Higher cultivatable fungal load and frequency in the gastrointestinal tract of BD donors by cultivation. The CFU/mL was measured for luminal samples (A) and mucosal samples (B) in the stomach, duodenum, jejunum, ileum, and colon after 24 hour incubations on YPD media agar plates. Each donor is represented by a unique shape as indicated in the legend (BD: n = 6 and DCD: n = 5) with a 10 CFU/mL limit of detection. Mann-Whitney *U* test statistical significance with Holm-Šidák correction cutoff of *P* < 0.05. Exact *P* values for luminal (A) samples are *P* = 0.040, *P* = 0.012, *P* = 0.007, *P* = 0.600, and *P* = 0.005 for the stomach, duodenum, jejunum, ileum, and colon, respectively. For mucosal samples (B), *P* = 0.012, *P* = 0.004, *P* = 0.002, *P* = 0.597, and *P* = 0.001, for stomach, duodenum, jejunum, ileum, and colon, respectively. *P<0.05 and **P< 0.01

TABLE 5 Few fungal isolates have reduced inhibition to multiple classes of antifungal drugs *in vitro*[a]

| | Amphotericin B (µg/mL) | Caspofungin (≥1 and >0.5 µg/mL) | Micafungin (≥1 and ≥0.25 µg/mL) | Fluconazole (≥8 and ≥64 µg/mL) |
|---|---|---|---|---|
| *C. tropicalis,* donor 1 | 0.13 | S | S | R* |
| *N. glabratus,* donor 1 | 0.31 | S | S | R* |
| *N. glabratus,* donor 6 | 0.27 | R* | S | R* |
| *N. glabratus,* donor 2 | 0.42 | S | S | R* |
| *C. albicans-A,* donor 8 | 0.24 | S | S | R* |
| *C. albicans-B,* donor 8 | 0.07 | S | S | R* |
| *C. albicans-A,* donor 9 | 0.08 | S | S | R* |
| *C. albicans-B,* donor 9 | 0.06 | S | S | R* |
| *C. albicans,* donor 10 | 0.10 | S | S | R* |
| *C. albicans* SC5314 | 0.13 | S | S | R* |

[a]Representative fungal isolates were tested for minimum inhibitory concentrations to common antifungals from MIC test strips. Breakpoints were determined according to the Clinical and Laboratory Standards Institute. R* indicates resistance, and S indicates susceptibility. Amphotericin B does not have official breakpoints and was reported as average concentrations ($n = 3$).

the genome of clinical isolates, the variability of *HWP1* and *AWP1* suggests potential commensal-like adaptation to the intestinal environment (39). Overall, the majority of fungal isolates harbored reported virulence genes, and future work could determine if the isolates exhibit pathogenicity *in vivo*.

## DISCUSSION

Here, we report differences in fungal burden in the GIT of BD and DCD organ donors. Donor care prior to procurement differed between the donor types, specifically in nutrient delivery and administration of corticosteroids. We found up to 10,000 times higher fungal burden in the GIT of BD donors compared to DCD by culture-independent and dependent methods. Furthermore, all tested GI *Candida* spp. and *N. glabratus* isolates were resistant to fluconazole and encoded expected virulence genes. We hypothesize that nutrient delivery and corticosteroid usage may alter the GIT environment to be conducive to fungal overgrowth. However, we recognize our smaller sample size as a limitation to this study. Due to the small size, we did not detect any IFI in the recipients of the donors in our study, emphasizing the need for future work to increase in sample size.

While IFIs are a risk factor for solid organ transplant recipients, the source of the fungi causing the infection is most often undetermined. *Candida* spp. and *N. glabratus* are opportunistic pathogens responsible for systemic blood infections, endocarditis, soft tissue infections, and mucosal site infections (40). Furthermore, *Candida* spp. are the most frequent cause of infections in SOT across all transplanted organs except the lungs (5). Specifically in small intestinal transplants, *Candida* spp. account for up to 85% of IFIs (3). Therefore, the GIT may be a risk factor that should be further investigated as a source of IFI in organ donation.

Antifungal resistance is considered a serious threat by the World Health Organization (7). Of the isolates we purified from the GIT, we found all tested isolates were resistant to

fluconazole, and one isolate was resistant to an echinocandin (caspofungin). Echinocandin resistance is often associated with hotspot mutations in the *FKS1* gene, but we did not find any evidence of those hotspot mutations in the resistant isolate (41). Our observed increased flucanzole resistance correlated with *FLU1*, *ADA2*, *SNQ2*, and *CDR1* presence, which have been associated with azole resistances (21–23, 25). Further *in vivo* testing of antifungal resistance is required to determine if the isolates remain resistant when maneuvering the host immune system. However, the presence of resistant fungi found in the GIT of these organ donors may raise concerns for IFI treatment in recipients and consideration for preventative measures in the donors.

Another factor that may contribute to fungal overgrowth is the exposure of the donor to antibiotic therapy. The gut microbiome is thought to help maintain the low fungal population through competition (42). In humans, oral antibiotic treatment is a risk factor for fungal infections (10, 43, 44). In this study, while we noted BD donors had greater IV antibiotic exposure, the amount of IV antibiotics that reached the GIT lumen is unknown. Despite the fact that BD donors were administered more antibiotics than DCD donors, we did not find a difference in bacterial burden across donors by RT-qPCR (Fig. S1C and D). While we cannot rule out any changes to the functionality of the gut microbiome due to IV antibiotics, we do not suspect they are the main driver behind the GIT fungal overgrowth.

We suggest that the main drivers of fungal overgrowth could be the differences in how BD and DCD donors were cared for before procurement. We observed that BD donors spent multiple days longer without enteral nutrition compared to DCD donors (Table 1). This would result in an absence of nutrients in the gut and reduced intestinal motility, which has been directly linked with small intestinal fungal overgrowth (45). This altered intestinal environment could directly impact fungal physiology. Previous work has shown that stress factors such as starvation, alterations in gut metabolites, shifts in pH, and the microbiome prompt fungi to switch to the pathogenetic and faster-growing pseudo-hyphal and hyphal forms of *Candida* spp. (46, 47). Furthermore, fungal overgrowth is associated with dysmotility in GIT diseases such as irritable bowel syndrome (48). The other potential contributor was the corticosteroid treatment that only BD donors recieved. Corticosteroids are a known risk factor for invasive candidemia through suppression of the immune system and have been reported to increase cases of IFI in people with inflammatory bowel disease (4, 49, 50). Therefore, the combination of reduced nutrients and slower GIT motility could create an environment favorable for fungal overgrowth in the GIT. If the GIT fungal overgrowth translocates systemically, the suppressed immune system, due to the corticosteroids, could allow for successful fungal translocation to other organs in the donor that get passed to the recipients.

Taken together, this unexpected finding warrants further investigation into the gastrointestinal tract of BD donors as a source of IFIs in transplant recipients. In future research, we aim to increase our sample size to provide a more comprehensive assessment of the GIT fungal burden of organ donors. Furthermore, future work would include altering steroid usage and enteral feeding in potentially preventing and combating IFIs in transplant recipients.

## ACKNOWLEDGMENTS

The researchers wish to express their sincere gratitude to the donor families for their gift to support the advancement of science and the betterment of healthcare. We would also like to thank everyone in the Britton lab who has helped collect samples and LifeGift for making this project possible. A special acknowledgment to Katherine Wozniak for helpful feedback on this manuscript. We would like to acknowledge David Corry and Kelsey Mauk for their help in providing C. albicans 5314 and feedback on the manuscript (K.M.). We would also like to acknowledge the work completed by Mathew Ross and Juwan Cormier at the CMMR for their help in running the FungiQuant assays and project management.

This work has been funded by Baylor College of Medicine.

E.J.N. contributed to concept/design, data analysis/interpretation, drafting the article, critical revision of the article, approval of the article, and statistics. A.K.A. performed fungal data analysis, drafted the article, and helped with approval of the article. C.K.A., M.M.B., C.M.R., M.N.S., N.M.V., Z.L., E.C.P., H.A.D., S.C.D.R., and J.N. contributed to data collection, concept/design, critical review of the article, and approval of the article. Y.T.B. interpreted the data and also contributed to critical review and paper approval. R.A.B. contributed to concept/design, drafting, critical review, approval of the article, and funding acquisition.

## AUTHOR AFFILIATIONS

[1]LifeGift, Houston, Texas, USA
[2]Department of Molecular Virology and Microbiology, Baylor College of Medicine, Houston, Texas, USA

## PRESENT ADDRESS

Madeline M. Bresson, Department of Pathology, Microbiology, and Immunology, Vanderbilt University Medical Center, Nashville, Tennessee, USA
Nora M. Villafuerte, Department of Biological Sciences, Louisiana State University, Baton Rouge, Louisiana, USA
Zhe Lyu, Department of Plant and Microbial Biology, NC State University, Raleigh, North Carolina, USA

## AUTHOR ORCIDs

Erika J. Nachman  http://orcid.org/0000-0002-0897-2475
Zhe Lyu  http://orcid.org/0000-0003-0184-6609
Yolanda T. Becker  http://orcid.org/0009-0000-7053-0435
Robert A. Britton  http://orcid.org/0000-0001-8983-9539

## AUTHOR CONTRIBUTIONS

Erika J. Nachman, Resources, Supervision, Writing – review and editing, Conceptualization, Resources, Investigation, Writing – review and editing | Colleen K. Ardis, Conceptualization, Writing – original draft, Resources | A. Kyle B. Ardis, Supervision, Project administration | Javier Nieto, Resources | Madeline M. Bresson, Resources | Clare M. Robertson, Resources | Maggie N. Seale, Resources | Nora M. Villafuerte, Resources | Zhe Lyu, Resources | Eva C. Preisner, Writing – review and editing, Resources | Heather A. Danhof, Writing – review and editing, Resources, Writing – review and editing | Sara C. Di Rienzi, Resources, Writing – review and editing, Resources, Writing – review and editing | Yolanda T. Becker, Resources, Supervision, Writing – review and editing | Robert A. Britton, Resources, Project administration, Funding acquisition, Supervision, Investigation, Writing – review and editing

## DATA AVAILABILITY

Raw sequencing data and genome assembly an be found at NCBI PRJNA1170939. R scripts can be found at https://github.com/enachman/highfungiGI_data/tree/main.

## ETHICS APPROVAL

In collaboration with LifeGift, a local organ procurement organization, the legal next of kin authorized the recovery of the GIT for research and education for all GIT included in this study. This study had an IRB exemption, and patient information was de-identified.

## ADDITIONAL FILES

The following material is available online.

## Supplemental Material

**Figure S1 (Spectrum03341-24-s0001.tiff).** qPCR data for fungal load.
**Figure S2 (Spectrum03341-24-s0002.tif).** Presence of anti-fungal genes in isolates.
**Figure S3 (Spectrum03341-24-s0003.tiff).** Presence of virulence genes in isolates.
**Supplemental materials (Spectrum03341-24-s0004.docx).** Tables S1 to S4, supplemental figure legends, and additional methods.

## Open Peer Review

**PEER REVIEW HISTORY (review-history.pdf).** An accounting of the reviewer comments and feedback.

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
