## [Reviewer comments · Microbiology Spectrum]

Microbiology Spectrum

Increased Fungal Burden in the Gastrointestinal Tract of Brain-Dead Organ Donors

Erika Nachman, Colleen Ardis, Andrew Ardis, Javier Nieto, Madeline Bresson, Clare Robertson, Maggie Seale, Nora Villafuerte, Zhe LYU, Eva Preisner, Heather Danhof, Sara Dirienzi, Yolanda Becker, and Robert Britton

Corresponding Author(s): Robert Britton, Baylor College of Medicine

Review Timeline:

Submission Date:	January 2, 2025
Editorial Decision:	February 17, 2025
Revision Received:	April 8, 2025
Editorial Decision:	April 23, 2025
Revision Received:	April 28, 2025
Accepted:	April 29, 2025

Editor: James Konopka

Reviewer(s): Disclosure of reviewer identity is with reference to reviewer comments included in decision letter(s). The following individuals involved in review of your submission have agreed to reveal their identity: Cornelius J. Clancy (Reviewer #2)

Transaction Report:

DOI: <https://doi.org/10.1128/spectrum.03341-24>

Re: Spectrum03341-24 (Increased Fungal Burden in the Gastrointestinal Tract of Brain-Dead Organ Donors)

Dear Dr. Robert A Britton:

Thank you for the privilege of reviewing your work. Below you will find my comments, instructions from the Spectrum editorial office, and the reviewer comments.

The reviewers agreed that your manuscript addressed an interesting topic. However, they had some concerns that you should address.

Revision Guidelines

Sincerely,
James Konopka
Editor
Microbiology Spectrum

Reviewer #1 (Comments for the Author):

In the manuscript "Increasing Fungal Burden in the Gastrointestinal Tract of Brain-Dead Organ Donors" by Nachman et al., authors examine the mycobiome of both brain-dead and circulatory-dead organ donors. While the concept of the manuscript is worthy of investigation, I have a few questions and comments that should be addressed before publication.

Major:

- Overall it is hard to analyze the impact of these results. First, it is a very small amount of patients analyzed so even if the statistical test says it's significant, it's still hard to assume it would hold up in a larger population. Second, it is well established that fungi are routinely found colonizing the GI tract and their significance is questionable. It has been found that the abundance and composition of the mycobiome is affected by age, sex, diet, antibiotic and steroid administration, and a number of other factors that are likely confounding variables between your 2 groups. As such, it is not surprising that the BD donors would have a higher fungal burden than the DCD donors since they were treated with more antibiotics and were given steroids. Even if true, authors have not provided evidence that the higher burden translates to higher rates of invasive infection. Have there been studies that show transplant patients who receive organs from BD donors have higher rates of IFIs? Can you look at outcomes of the patients who received organs from the donors you assessed? Is there an association between IFI and specific organs that are transplanted?

-Why were specimens processed anaerobically? Specimens should not be processed in such a way if you want to recover fungal organisms. This likely severely reduced your ability to recover viable organisms. This seems to be the case since you mention you only recovered 28 isolates total.

-If trying to assess burden, why were luminal samples diluted based on weight? It would make more sense to homogenize in a uniform volume so that you can accurately compare between the patients like you did for mucosal specimens. Supplementary figure 1 is the most convincing figure because gene copies are normalized to overall DNA amount and results are not reliant on being able to recover viable fungi. This should be a main figure and used to make your point, not the CFU data because that is likely heavily biased by the anaerobic processing mentioned above.

-If able, it would be interesting to look at the complete mycobiome composition genetically to see if there are overall differences between the 2 donor types. Some species are more prone to invasion than others so it would be good to see all the species found in the mycobiome, not just the ones you happened to recover (which as discussed before, may have been affected by the anaerobic treatment). This is also true for looking at virulence genes since your current analysis is only on isolates that happened to be recovered in culture.

-Please note that different species have different breakpoints and if you want to label an isolate as susceptible or resistant, they should be tested following CLSI guidelines. As presented, using a cutoff of 2ug/mL of caspofungin is not accurate. The breakpoint for resistance for *C. albicans* is {greater than or equal to} 1 so if an isolate in your study has an MIC of 1, by your standards it is labelled as susceptible, but by CLSI and FDA guidelines, it is actually resistant. The same is true for micafungin. Last, there are no breakpoints for amphotericin B. Though there is a suggested breakpoint from the CDC of 2ug/mL for *Candida auris*, this does not translate to other yeast species and is not a validated breakpoint. That should be clarified in the manuscript.

Minor:

-In the abstract, lines 50 and 59 are discrepant. Were BD donors parenterally or enterally fed?

-Donor Inclusion/Exclusion Criteria: What was the cutoff time for circulatory death?

-Line 216: Fungi were only isolated from 2 DCD donors? Is this a typo?

-WGS analysis: If you are analyzing pure isolates, why was contamination set all the way up to 30% and completeness set down to 50%?

-Table 1: P values should be shown for differences in all categories. It would not be surprising if each of these categories were confounding.

-Table 2: Keep consistent. Rocephin is ceftriaxone. Might also be good to stick to the generic names, and not the brand names (i.e Zosyn should be Piperacillin / Tazobactam)

Reviewer #2 (Comments for the Author):

This is a well-written and useful report of increased fungal burdens in the GI tracts (GITs) of brain dead (BD) organ donors compared to donation after circulatory death (DCD) donors. The study is the first to quantify fungi from donor GIT by both culture-dependent and genome-based methods. Methods are appropriate (and this section of the paper is very clearly written), as are conclusions drawn from the data. I would have liked to see more strain phenotypes beyond the breakpoint MIC assays for certain antifungals. However, the authors acknowledge such assays as a topic for future study, which I think is acceptable. I have several relatively minor comments for the authors' consideration.

1. In reading the Abstract, the sentence on lines 54-55 is not particularly clear. Where were these spp. detected from? How many patients? Tweaking language would help readers.

2. Was there no information on causes of death?

3. What was the time period for the antibiotic and antifungal treatment data in donors? Do the data reflect all information available from the hospitalization that culminated in death?

4. Why wouldn't strains contain virulence genes? I'm surprised the % detected for several genes to be surprising low. Can the authors briefly discuss or speculate why so many of these are missing in many strains?

5. How were the 23 isolates selected for WGSing?
6. Were FKS mutations evident in strains with diminished echinocandin susceptibility (or in any strains, for that matter)?
7. I also was surprised to see that all strains were fluconazole resistant. Any thoughts or speculation on why this may be?
8. 6/6 BD and 3/5 DCD donors received an antibiotic with anaerobic activity. I might have expected anaerobic coverage in particular to impact fungal populations, which was not the case. This point might merit mentioning in discussion.

Dear Editor and Reviewers,

Thank you for your insightful comments on our manuscript "Increased Fungal Burden in the Gastrointestinal Tract of Brain-Dead Organ Donors" (ID:Spectrum03341-24). The manuscript has been edited based on the reviews as well as new data to better support the conclusions as requested. Below, please find our point-by-point response to the comments from the reviewers.

Response to Reviewer #1

Major Comment 1: *First, it is a very small amount of patients analyzed so even if the statistical test says it's significant, it's still hard to assume it would hold up in a larger population. Second, it is well established that fungi are routinely found colonizing the GI tract and their significance is questionable. It has been found that the abundance and composition of the mycobiome is affected by age, sex, diet, antibiotic and steroid administration, and a number of other factors that are likely confounding variables between your 2 groups. As such, it is not surprising that the BD donors would have a higher fungal burden than the DCD donors since they were treated with more antibiotics and were given steroids. Even if true, authors have not provided evidence that the higher burden translates to higher rates of invasive infection*

Response: While we recognize that this study size is small, we, nevertheless, found statistical significance in the frequency and concentration of fungal load between BD and DCD donors. It remains to be seen if this trend is seen in larger populations. We hope these results bring awareness of potential fungal overgrowth in organ donation and encourage further research into comparing BD and DCD donor mycobiomes to improve organ health and recipient outcomes. Secondly, the SI mycobiome is largely unexplored, and it is unclear if fungi are frequently detected in the small intestine in healthy adults. It is documented from stool samples to be a small component of the gut microbiome, so our findings highlight this unexpected result in an organ donation context.

Major Comment 2: *Have there been studies that show transplant patients who receive organs from BD donors have higher rates of IFIs? Can you look at outcomes of the patients who received organs from the donors you assessed? Is there an association between IFI and specific organs that are transplanted?*

Response: There is an association between organ type and rate of IFI infection. The small bowel has the highest rate of infection at 11.6%. Followed by lung, liver, heart, pancreas, and kidney (8.6%, 4.7%, 4.0%, 3.4%, and 1.3%)[Shoham]. We hypothesized that one reason why small bowel transplants have the highest rate of IFI could be from previously undetected fungal overgrowths in the intestine. The statistic that IFI occurs in 2.5-3% of organ donors comes from a study that surveyed organ donations between 2001-2006 in the USA where they only had access to BD donor data as DCD donations were not legal in the US until 2007 [Pappas]. While there has not been a study that compared equal groups of BD and DCD donors for IFI, there have been a few that have studied IFI in recipients of DCD organs. One study found the rate of IFI after kidney donation from DCD donors was 1/78 (1.2%). This resulted in 2 IFI in the recipients. The other found 1 DCD donor out of 17, resulting in *C. albicans* in the bloodstream of the donor. This did not result in any infection in the recipients[Ye, World of Gastroenterology, 2017]. Overall, there have not been any studies to date that directly compare the mycobiome of two organ donor types, which emphasizes the novelty of this study and what we begin to address. We did investigate the outcomes of the patients from our study and did not find any IFI in the recipients from this study probably due to our smaller sample size. In future work, a larger sample size would be necessary to assess the risk of IFI from organ donor types.

Major Comment 3: *Why were specimens processed anaerobically? Specimens should not be processed in such a way if you want to recover fungal organisms. This likely severely reduced your ability to recover viable organisms. This seems to be the case since you mention you only recovered 28 isolates total.*

Response: The samples were processed anaerobically to maintain the anaerobic environment of the intestinal tract. However, the samples were only exposed to anaerobic conditions for a maximum of 1 hour during our processing step. Culturing the processed samples and *in vitro* testing of the fungal isolates was performed in aerobic conditions. This resulted in thousands of yeasts from the cultivation of the BD donors as seen in Figure 1. We highlighted 23 isolates as a small subset to gain further insight into the fungal composition and potential pathogenicity.

Major Comment 4: *If trying to assess burden, why were luminal samples diluted based on weight? It would make more sense to homogenize in a uniform volume so that you can accurately compare between the patients like you did for mucosal specimens. Supplementary figure 1 is the most convincing figure because gene copies are normalized to the overall DNA amount and results are not reliant on being able to recover viable fungi. This should be a main figure and used to make your point, not the CFU data because that is likely heavily biased by the anaerobic processing mentioned above.*

Response: The methods have been updated to accurately reflect that we measured weight for both luminal and mucosal samples to make equal comparisons across organs and donors. We chose to make the qPCR data a supplemental figure because we believe viable microbes are a more reliable way to estimate the fungal load in the intestine. While mucosal samples were statistically significant, many luminal BD samples had very low ITS2 signals. qPCR methods especially from typically low biomass site such as the small intestine are prone to excess host contamination or bacterial DNA or luminal metabolites that could have made it difficult to detect fungi in those samples. Nevertheless, trends are the same between both quantification methods where there is a higher fungal burden in the BD donors.

Major Comment 5: *If able, it would be interesting to look at the complete mycobiome composition genetically to see if there are overall differences between the 2 donor types. Some species are more prone to invasion than others so it would be good to see all the species found in the mycobiome, not just the ones you happened to recover (which as discussed before, may have been affected by the anaerobic treatment). This is also true for looking at virulence genes since your current analysis is only on isolates that happened to be recovered in culture.*

Response: We were unable to perform complete mycobiome sequencing to analyze the genetic composition due to the low biomass of the samples.

Major Comment 6: *Please note that different species have different breakpoints and if you want to label an isolate as susceptible or resistant, they should be tested following CLSI guidelines. As presented, using a cutoff of 2ug/mL of caspofungin is not accurate. The breakpoint for resistance for *C. albicans* is {greater than or equal to} 1 so if an isolate in your study has an MIC of 1, by your standards it is labelled as susceptible, but by CLSI and FDA guidelines, it is actually resistant. The same is true for micafungin. Last, there are no breakpoints for amphotericin B. Though there is a suggested breakpoint from the CDC of 2ug/mL for *Candida auris*, this does not translate to other yeast species and is not a validated breakpoint. That should be clarified in the manuscript.*

Response: These breakpoints have been updated according to each species tested. To further follow the guidelines, the antibiotics resistance test was redone on RPMI medium with 2% glucose and MOPS using E-STRIPs to best determine the breakpoints of each of the strains tested. We also included *C. albicans* SC5314 as a control instead of *S. cerevisiae* to better compare expected resistance profiles.

Minor Comment 1: *In the abstract, lines 50 and 59 are discrepant. Were BD donors parenterally or enterally fed?*

Response: They were parenterally fed. This has been updated in the text.

Minor Comment 2: *Donor Inclusion/Exclusion Criteria: What was the cutoff time for circulatory death?*

Response: The cut-off time was up to 90 minutes after the removal of life-support machinery.

Minor Comment 3: *Line 216 Fungi were only isolated from 2 DCD donors? Is this a typo?*

Response: This was not a typo. We only consistently found fungi from 2 DCD donors, donor 6 and 10. The original paper included 1 point for Donor 5 which was lower than the detection limit of the assay that has been removed.

Minor comment 4: WGS analysis: *If you are analyzing pure isolates, why was contamination set all the way up to 30% and completeness set down to 50%?*

Response: After further investigation into this comment, we decided to change the assembly method of the fungi isolates from metagenome-assembled genomes to whole genome assembly. The genomes were reassembled using SPAdes because it is optimized for short-length paired-end reads and has been used to assemble other fungal genomes in the field. When compared to reference genomes using Companion, the new assemblies achieved contained 95% to 100% compared to the reference genomes [William Haese-Hill, Kathryn Crouch, Thomas D. Otto Annotation and visualization of parasite, fungi and arthropod genomes with Companion. *Nucleic Acids Research*, 2024; gkae378. DOI: [10.1093/nar/gkae378](https://doi.org/10.1093/nar/gkae378)]. Isolates were dropped that had higher contamination.

Minor Comment 5: *Table 1: P values should be shown for differences in all categories. It would not be surprising if each of these categories were confounding.*

Response: There was no significant difference between age in BD and DCD (Mann-U, $p=.2332$). There was no significant difference between BD and DCD in sex and ethnicity as well (Fishers Exact Test, $p=0.56$, $p=0.06$, respectively). The ethnicity p-value is close to significance, so it was subcategorized into Hispanic and non-Hispanic (Fishers Exact Test, $p=0.06$). These p-values have been updated in the text.

Minor Comment 6: Keep consistent. Rocephin is ceftriaxone. Might also be good to stick to the generic names, and not the brand names (i.e Zosyn should be Piperacillin / Tazobactam)

Response: This has been fixed in Table 2.

Response to Reviewer #2

Comment 1: *In reading the Abstract, the sentence on lines 54-55 is not particularly clear. Where were these spp. detected from? How many patients? Tweaking language would help readers.*

Response: Sentences on lines 54-55 have been adjusted for clarity.

Comment 2: *Was there no information on causes of death?*

Response: There is information cause of death and has been added as a table in the supplementary information.

Comment 3: *What was the time period for the antibiotic and antifungal treatment data in donors? Do the data reflect all information available from the hospitalization that culminated in death?*

Response: The time period for the antibiotics and antifungals are listed in Table 1 from their intake until procurement. This has been confirmed on the clinical side and accurately reflects all information prior to procurement.

Comment 4: *Why wouldn't strains contain virulence genes? I'm surprised the % detected for several genes to be surprising low. Can the authors briefly discuss or speculate why so many of these are missing in many strains?*

Response: After genome reassembly of the genomes using SPAdes, the genes that are supposed to be present are now found in every genome. There is still variability in the *ALS1* gene where half the isolates are null. However, all isolates have *ALS3*, so variability could suggest redundant function across the ALS adhesion family.

Comment 5: *How were the 23 isolates selected for WGSing?*

Response: The priority was to include at least one strain from any donor with a positive fungal signal. I next chose multiple isolates of BD donors 1, 3, 8, and 9 because they had consistent and high fungal signals. Donors 4 and 13 either had little signal (donor 4) or less consistent across the gut (donor 13). Lastly, I included two isolates from each DCD donor that had a positive fungal signal.

Comment 6: *Were FKS mutations evident in strains with diminished echinocandin susceptibility (or in any strains, for that matter)?*

Response: I aligned the SPAdes assembled genomes to the *FKS1* gene to look for the known, published hotspots in *FKS1* that have been associated with diminished echinocandin susceptibility (641-FLTSLRDP-649; hot spot 2, 1357-DWIRRYTL-1364). I did not detect any mutations in those hotspots for any isolates. I did detect other mutations (G9E, P1837A) in *C. albicans* donor 8B, 9A, and 9B. *C. albicans* donor 10 had only a P1837A. Therefore, the known hotspots are not the cause of increased resistance in the strains, and these mutations in the beginning and ends of the peptide could be novel hotspots and affect the susceptibility to echinocandins.

Comment 7: *I also was surprised to see that all strains were fluconazole resistant. Any thoughts or speculation on why this may be?*

Response: We found fluconazole-related resistance genes *FLU1* and *FCR1* in 100% of *C. albicans*. While *N. glabratus* and *C. tropicalis* isolates only had *FLU1*, those isolates also contained other multi-drug resistance-related genes, *ADA2* and *CDR1*. As a control, we also found the wild-type *C. albicans* strain SC5314 was resistant to fluconazole. Clinical data suggests that up to 35% of *Candida albicans* isolates from *Candida* vaginitis were resistant to fluconazole[Sobel 2023]. There could also be discrepancies between *in vitro* testing and *in vivo* treatment, such as the impacts of host immune response and bacterial competition that allows for increased resistance without external pressures.

Comment 8: 8. 6/6 BD and 3/5 DCD donors received an antibiotic with anaerobic activity. I might have expected anaerobic coverage in particular to impact fungal populations, which was not the case. This point might merit mentioning in discussion.

Response: While we do think antibiotics may have played a minor role, they were administered via I.V, and so it is also hard to quantify how much and how evenly the antibiotics penetrated through the lumen and acted on the microbial populations (lines431-439). Based on our 16S qPCR data in supplementary figure 1, there does not appear to be a difference in microbial load between the donor types, further supporting why we do not believe this is a driving factor for the fungal overgrowth. Additionally, we would expect anaerobic antibiotics to impact the small and large intestines differently due to the higher facultative anaerobes in the small intestine.

Re: Spectrum03341-24R1 (Increased Fungal Burden in the Gastrointestinal Tract of Brain-Dead Organ Donors)

Dear Dr. Robert A Britton:

Thank you for the privilege of reviewing your work. Below you will find my comments, instructions from the Spectrum editorial office, and the reviewer comments.

Reviewer 1 thought your revisions were very helpful and greatly improved your manuscript. They have just a few minor comments to address. Reviewer 2 is not responding to a request to review your revised manuscript, so I am moving forward with the comments from Reviewer 1. However, I think it would be helpful to put some of your replies to Reviewer 2 into the revised manuscript. On a personal note, I thought this was an interesting study and will look forward to reading about your research in the future.

Revision Guidelines

Sincerely,
James Konopka
Editor
Microbiology Spectrum

Reviewer #1 (Comments for the Author):

Thank you for your thorough responses and edits to our previous comments. They were helpful in better understanding the

background of the study and what was done. As such, I think more of these responses should be included in the manuscript and not just for the reviewers.

Specifically:

- The detail in the response to Major comment 2 provides added context for the study and would be useful in either the intro or discussion. This could be further expanded in the discussion where authors can highlight the limitations of this study and propose the future studies mentioned in the response.
- The detailed response to comment 5 is also useful information that should be included in the methods or results section so readers can understand how the isolates were picked.

Also, I noticed the denominators in tables 3 and 4 were not fully described. In the methods it says each patient had 13 sections that were analyzed (so 65 from DCD and 91 from BD) but table 3 has 90 and 105, respectively. However, based on the description for Table 3 that states some, but not all patients were done in duplicates. Authors should normalize these findings as this will bias the results (patient's who were run in duplicates will outweigh those who weren't) and this should be described in the methods.

Minor comments:

-Table 1 should be described as demographics, not metadata

-Consider rephrasing lines 400-401 in "fungal isolates are resistant to antifungals". Using "had" makes it sound like those patients only had those isolates, not that only a select few were analyzed. This can be fixed if you include the detailed response to comment 5 as mentioned above.

-Double check table 6 description and column headers. They do not reflect the updated methodologies.

-Sup fig 1. Add ITS gene name into description to detail what was quantified

Dear Editor and Reviewers,

Thank you for your insightful comments and continued interest in our manuscript “Increased Fungal Burden in the Gastrointestinal Tract of Brain-Dead Organ Donors” (ID:Spectrum03341-24). The manuscript has been edited based on the reviews as well as new data to better support the conclusions as requested. All line references are to the clean manuscript. Below, please find our point-by-point response to the comments from the reviewer.

Revision #2

Response to Reviewer #1

Major Comment 1: *The detail in the response to Major comment 2 provides added context for the study and would be useful in either the intro or discussion. This could be further expanded in the discussion where authors can highlight the limitations of this study and propose the future studies mentioned in the response.*

Response: The single DCD IFI study has been added in the introduction (line 107) to add better context to the field. We have also commented on the limitations of the study and propose future directions from the response in the discussion in lines (405-408, 460-461).

Major Comment 2: *The detailed response to comment 5 is also useful information that should be included in the methods or results section so readers can understand how the isolates were picked.*

Response: This has been updated to include the response to reviewer 2's comment 5 on the priority of strains to be WGS in the results section (346-352).

Major Comment 3: *Also, I noticed the denominators in tables 3 and 4 were not fully described. In the methods it says each patient had 13 sections that were analyzed (so 65 from DCD and 91 from BD) but table 3 has 90 and 105, respectively. However, based on the description for Table 3 that states some, but not all patients were done in duplicates. Authors should normalize these findings as this will bias the results (patients who were run in duplicates will outweigh those who weren't) and this should be described in the methods.*

Response: This should be clarified better--while our sample collection protocol aimed for 13 sites, 5 sites were selected to represent each section of the intestinal tract that were cultivated (Table 2, Table 3, and Figure 2) and sequenced (SF1). The selected regions were the fundus, duodenum, middle jejunum, middle ileum, and ascending colon. The maximum number of samples possible are 120 for BD (5 sites, 6 donors, duplicates) or 100 for DCD (5 sites, 5 donors, duplicates). We have 104 samples from BD (52 luminal, 52 mucosal) and 90 from DCD (48 luminal, 42 mucosal). This discrepancy is because we did not have the ascending luminal colon samples from donor 2 (BD), donor 8 (BD), donor 10(DCD), and donor 13(BD). From the mucosal side, we did not have the stomach, duodenum, jejunum, and colon from donor 9(BD), the fundus from donor 6(DCD), and the stomach, jejunum, and colon from donor 10(DCD). All samples were completed in duplicates, except LG13 colon only had 1 replicate, and so LG13 colon was removed from the analysis to keep consistency. The methods (216-222) have been updated to clarify the source of the data, and statistics were re-run to confirm that the data is still statistically significant.

Minor Comment 1: *Table 1 should be described as demographics, not metadata*

Response: “Metadata” has been changed to demographics Table 1 and Supplemental Table 1.

Minor Comment 2: *Consider rephrasing lines 400-401 in "fungal isolates are resistant to antifungals". Using "had" makes it sound like those patients only had those isolates, not that only a select few were analyzed. This can be fixed if you include the detailed response to comment 5 as mentioned above.*

Response: Comment 5# has been added with the addition of highlighting that these are a selective group rather than all the isolates present (346-352).

Minor Comment 3: *Double check table 6 description and column headers. They do not reflect the updated methodologies.*

Response: This has been double-checked for Table 5 (there is no Table 6). The methods (using MIC strips on RPMI plates,(674-679) should accurately reflect the methodology (268-276)

Minor Comment 4: *Sup fig 1. Add ITS gene name into description to detail what was quantified*

Response: This has been updated to note the ITS2 gene.

Response to Editor

Major Comment 1: *However, I think it would be helpful to put some of your replies to Reviewer 2 into the revised manuscript.*

Response: We have incorporated the insightful comments from comments 4, 5, 6, 7, and 8 from reviewer #2, strengthening the paper. Comment #5 from reviewer #2 has been included (346-352), explaining the reasoning behind the strain choice for WGS. We also included notes from comments 6 and 7 referring to fluconazole resistance (423-424) and FKS1 mutations (421-424), noting that the strains would be expected to have most of the virulence genes present and speculation on the increased fluconazole resistance. We also comment on the varied *ALS1* presence derived from comment 4; it has been found that clinical *Candida* isolates can fluctuate in *ALS1* presence (393-395), supporting our results in *ALS1* presence diversity. Lastly, we addressed comment 8 with our stance that antibiotics likely only played a small role in the fungal overgrowth as we did not find any significant changes in 16S rRNA qPCR between donor types (SF1) (332-340).

Revision #1

Response to Reviewer #1

Major Comment 1: *First, it is a very small amount of patients analyzed so even if the statistical test says it's significant, it's still hard to assume it would hold up in a larger population. Second, it is well established that fungi are routinely found colonizing the GI tract and their significance is questionable. It has been found that the abundance and composition of the mycobiome is affected by age, sex, diet, antibiotic and steroid administration, and a number of other factors that are likely confounding variables between your 2 groups. As such, it is not surprising that the BD donors would have a higher fungal burden than the DCD donors since they were treated with more antibiotics and were given steroids. Even if true, authors have not provided evidence that the higher burden translates to higher rates of invasive infection*

Response: While we recognize that this study size is small, we, nevertheless, found statistical significance in the frequency and concentration of fungal load between BD and DCD donors. It remains to be seen if this trend is seen in larger populations. We hope these results bring awareness of potential fungal overgrowth in organ donation and encourage further research into comparing BD and DCD donor mycobiomes to improve organ health and recipient outcomes. Secondly, the SI mycobiome is largely unexplored, and it is unclear if fungi are frequently detected in the small intestine in healthy adults. It is documented from stool samples to be a small component of the gut microbiome, so our findings highlight this unexpected result in an organ donation context.

Major Comment 2: *Have there been studies that show transplant patients who receive organs from BD donors have higher rates of IFIs? Can you look at outcomes of the patients who received organs from the donors you assessed? Is there an association between IFI and specific organs that are transplanted?*

Response: There is an association between organ type and rate of IFI infection. The small bowel has the highest rate of infection at 11.6%. Followed by lung, liver, heart, pancreas, and kidney (8.6%, 4.7%, 4.0%, 3.4%, and 1.3%)[Shoham]. We hypothesized that one reason why small bowel transplants have the highest rate of IFI could be from previously undetected fungal overgrowths in the intestine. The statistic that IFI occurs in 2.5-3% of organ donors comes from a study that surveyed organ donations between 2001-2006 in the USA where they only had access to BD donor data as DCD donations were not legal in the US until 2007 [Pappas]. While there has not been a study that compared equal groups of BD and DCD donors for IFI, there have been a few that have studied IFI in recipients of DCD organs. One study found the rate of IFI after kidney donation from DCD donors was 1/78 (1.2%). This resulted in 2 IFI in the recipients. The other found 1 DCD donor out of 17, resulting in *C. albicans* in the bloodstream of the donor. This did not result in any infection in the recipients[Ye, World of Gastroenterology, 2017]. Overall, there have not been any studies to date that directly compare the mycobiome of two organ donor types, which emphasizes the novelty of this study and what we begin to address. We did investigate the outcomes of the patients from our study and did not find any IFI in the recipients from this study probably due to our smaller sample size. In future work, a larger sample size would be necessary to assess the risk of IFI from organ donor types.

Major Comment 3: *Why were specimens processed anaerobically? Specimens should not be processed in such a way if you want to recover fungal organisms. This likely severely reduced your ability to recover viable organisms. This seems to be the case since you mention you only recovered 28 isolates total.*

Response: The samples were processed anaerobically to maintain the anaerobic environment of the intestinal tract. However, the samples were only exposed to anaerobic conditions for a maximum of 1 hour during our processing step. Culturing the processed samples and *in vitro*

testing of the fungal isolates was performed in aerobic conditions. This resulted in thousands of yeasts from the cultivation of the BD donors as seen in Figure 1. We highlighted 23 isolates as a small subset to gain further insight into the fungal composition and potential pathogenicity.

Major Comment 4: *If trying to assess burden, why were luminal samples diluted based on weight? It would make more sense to homogenize in a uniform volume so that you can accurately compare between the patients like you did for mucosal specimens. Supplementary figure 1 is the most convincing figure because gene copies are normalized to the overall DNA amount and results are not reliant on being able to recover viable fungi. This should be a main figure and used to make your point, not the CFU data because that is likely heavily biased by the anaerobic processing mentioned above.*

Response: The methods have been updated to accurately reflect that we measured weight for both luminal and mucosal samples to make equal comparisons across organs and donors. We chose to make the qPCR data a supplemental figure because we believe viable microbes are a more reliable way to estimate the fungal load in the intestine. While mucosal samples were statistically significant, many luminal BD samples had very low ITS2 signals. qPCR methods especially from typically low biomass site such as the small intestine are prone to excess host contamination or bacterial DNA or luminal metabolites that could have made it difficult to detect fungi in those samples. Nevertheless, trends are the same between both quantification methods where there is a higher fungal burden in the BD donors.

Major Comment 5: *If able, it would be interesting to look at the complete mycobiome composition genetically to see if there are overall differences between the 2 donor types. Some species are more prone to invasion than others so it would be good to see all the species found in the mycobiome, not just the ones you happened to recover (which as discussed before, may have been affected by the anaerobic treatment). This is also true for looking at virulence genes since your current analysis is only on isolates that happened to be recovered in culture.*

Response: We were unable to perform complete mycobiome sequencing to analyze the genetic composition due to the low biomass of the samples.

Major Comment 6: *Please note that different species have different breakpoints and if you want to label an isolate as susceptible or resistant, they should be tested following CLSI guidelines. As presented, using a cutoff of 2ug/mL of caspofungin is not accurate. The breakpoint for resistance for *C. albicans* is {greater than or equal to} 1 so if an isolate in your study has an MIC of 1, by your standards it is labelled as susceptible, but by CLSI and FDA guidelines, it is actually resistant. The same is true for micafungin. Last, there are no breakpoints for amphotericin B. Though there is a suggested breakpoint from the CDC of 2ug/mL for *Candida auris*, this does not translate to other yeast species and is not a validated breakpoint. That should be clarified in the manuscript.*

Response: These breakpoints have been updated according to each species tested. To further follow the guidelines, the antibiotics resistance test was redone on RPMI medium with 2% glucose and MOPS using E-STRIPS to best determine the breakpoints of each of the strains tested. We also included *C. albicans* SC5314 as a control instead of *S. cerevisiae* to better compare expected resistance profiles.

Minor Comment 1: *In the abstract, lines 50 and 59 are discrepant. Were BD donors parenterally or enterally fed?*

Response: They were parenterally fed. This has been updated in the text.

Minor Comment 2: *Donor Inclusion/Exclusion Criteria: What was the cutoff time for circulatory death?*

Response: The cut-off time was up to 90 minutes after the removal of life-support machinery.

Minor Comment 3: *Line 216 Fungi were only isolated from 2 DCD donors? Is this a typo?*

Response: This was not a typo. We only consistently found fungi from 2 DCD donors, donor 6 and 10. The original paper included 1 point for Donor 5 which was lower than the detection limit of the assay that has been removed.

Minor comment 4: WGS analysis: *If you are analyzing pure isolates, why was contamination set all the way up to 30% and completeness set down to 50%?*

Response: After further investigation into this comment, we decided to change the assembly method of the fungi isolates from metagenome-assembled genomes to whole genome assembly. The genomes were reassembled using SPAdes because it is optimized for short-length paired-end reads and has been used to assemble other fungal genomes in the field. When compared to reference genomes using Companion, the new assemblies achieved contained 95% to 100% compared to the reference genomes [William Haese-Hill, Kathryn Crouch, Thomas D. Otto Annotation and visualization of parasite, fungi and arthropod genomes with Companion. *Nucleic Acids Research*, 2024; gkae378.DOI: [10.1093/nar/gkae378](https://doi.org/10.1093/nar/gkae378)]. Isolates were dropped that had higher contamination.

Minor Comment 5: *Table 1: P values should be shown for differences in all categories. It would not be surprising if each of these categories were confounding.*

Response: There was no significant difference between age in BD and DCD (Mann-U, $p=.2332$). There was no significant difference between BD and DCD in sex and ethnicity as well (Fishers Exact Test, $p=0.56$, $p=0.06$, respectively). The ethnicity p-value is close to significance, so it was subcategorized into Hispanic and non-Hispanic (Fishers Exact Test, $p=0.06$). These p-values have been updated in the text.

Minor Comment 6: Keep consistent. Rocephin is ceftriaxone. Might also be good to stick to the generic names, and not the brand names (i.e Zosyn should be Piperacillin / Tazobactam)

Response: This has been fixed in Table 2.

Response to Reviewer #2

Comment 1: *In reading the Abstract, the sentence on lines 54-55 is not particularly clear. Where were these spp. detected from? How many patients? Tweaking language would help readers.*

Response: Sentences on lines 54-55 have been adjusted for clarity.

Comment 2: *Was there no information on causes of death?*

Response: There is information cause of death and has been added as a table in the supplementary information.

Comment 3: *What was the time period for the antibiotic and antifungal treatment data in donors? Do the data reflect all information available from the hospitalization that culminated in death?*

Response: The time period for the antibiotics and antifungals are listed in Table 1 from their intake until procurement. This has been confirmed on the clinical side and accurately reflects all information prior to procurement.

Comment 4: *Why wouldn't strains contain virulence genes? I'm surprised the % detected for several genes to be surprising low. Can the authors briefly discuss or speculate why so many of these are missing in many strains?*

Response: After genome reassembly of the genomes using SPAdes, the genes that are supposed to be present are now found in every genome. There is still variability in the *ALS1* gene where half the isolates are null. However, all isolates have *ALS3*, so variability could suggest redundant function across the ALS adhesion family.

Comment 5: *How were the 23 isolates selected for WGSing?*

Response: The priority was to include at least one strain from any donor with a positive fungal signal. I next chose multiple isolates of BD donors 1, 3, 8, and 9 because they had consistent and high fungal signals. Donors 4 and 13 either had little signal (donor 4) or less consistent across the gut (donor 13). Lastly, I included two isolates from each DCD donor that had a positive fungal signal.

Comment 6: *Were FKS mutations evident in strains with diminished echinocandin susceptibility (or in any strains, for that matter)?*

Response: I aligned the SPAdes assembled genomes to the *FKS1* gene to look for the known, published hotspots in *FKS1* that have been associated with diminished echinocandin susceptibility (641-FLTLRLDP-649; hot spot 2, 1357-DWIRRYTL-1364). I did not detect any mutations in those hotspots for any isolates. I did detect other mutations (G9E, P1837A) in *C. albicans* donor 8B, 9A, and 9B. *C. albicans* donor 10 had only a P1837A. Therefore, the known hotspots are not the cause of increased resistance in the strains, and these mutations in the beginning and ends of the peptide could be novel hotspots and affect the susceptibility to echinocandins.

Comment 7: *I also was surprised to see that all strains were fluconazole resistant. Any thoughts or speculation on why this may be?*

Response: We found fluconazole-related resistance genes *FLU1* and *FCR1* in 100% of *C. albicans*. While *N. glabratus* and *C. tropicalis* isolates only had *FLU1*, those isolates also contained other multi-drug resistance-related genes, *ADA2* and *CDR1*. As a control, we also found the wild-type *C. albicans* strain SC5314 was resistant to fluconazole. Clinical data suggests that up to 35% of *Candida albicans* isolates from *Candida* vaginitis were resistant to fluconazole [Sobel 2023]. There could also be discrepancies between *in vitro* testing and *in vivo* treatment, such as the impacts of host immune response and bacterial competition that allows for increased resistance without external pressures.

Comment 8: *8. 6/6 BD and 3/5 DCD donors received an antibiotic with anaerobic activity. I might have expected anaerobic coverage in particular to impact fungal populations, which was not the case. This point might merit mentioning in discussion.*

Response: While we do think antibiotics may have played a minor role, they were administered via I.V, and so it is also hard to quantify how much and how evenly the antibiotics penetrated through the lumen and acted on the microbial populations (lines 431-439). Based on our 16S qPCR data in supplementary figure 1, there does not appear to be a difference in microbial load between the donor types, further supporting why we do not believe this is a driving factor for the fungal overgrowth. Additionally, we would expect anaerobic antibiotics to impact the small and large intestines differently due to the higher facultative anaerobes in the small intestine.

Re: Spectrum03341-24R2 (Increased Fungal Burden in the Gastrointestinal Tract of Brain-Dead Organ Donors)

Dear Dr. Robert A Britton:

Your manuscript has been accepted, and I am forwarding it to the ASM production staff for publication. Your paper will first be checked to make sure all elements meet the technical requirements. ASM staff will contact you if anything needs to be revised before copyediting and production can begin. Otherwise, you will be notified when your proofs are ready to be viewed.

Sincerely,
James Konopka
Editor
Microbiology Spectrum